# Factors influencing the adoption of e-government by female university students

**Paula Andrea Rodríguez-Correa[1,2], Carlos Alberto Méndez-Rivera[2], Orfa Nidia Patiño-Toro[2], Alejandro Valencia-Arias [3]\*, Ada Lucia Gallegos Ruiz Conejo[4], Aarón Oré León[4], Jorge Tomás Cumpa Vásquez[3], Toño Eldrin Alvites Adan[3]**

1 Centro de Investigaciones, Institución Universitaria Escolme, Medellín, Colombia, 2 Facultad de Ciencias Económicas y Administrativas, Ingstituto Tecnológico Metropolitano, Medellín, Colombia, 3 Escuela de Ingeniería Industrial, Universidad Señor de Sipán, Chiclayo, Perú, 4 Instituto de Investigación y Estudios de la Mujer, Universidad Ricardo Palma, Lima, Perú

\* valenciajho@uss.edu.pe

## Abstract

The primary purpose of electronic government (e-government) is to promote transparency, facilitate access to government services, and strengthen the accountability of public institutions in the digital transformation age. However, few studies have explored the factors that affect women's adoption of e-government, especially in emerging economies. Consequently, this study aims to identify the factors that influence the adoption of e-government services by young women in Medellín, Colombia. To achieve this objective, a questionnaire was administered to a sample of 223 women, focusing on the factors proposed in the Technology Acceptance Model (TAM). The results were analyzed using Partial Least Squares Structural Equation Modeling (PLS-SEM). The findings confirmed the validity of both the measurement and structural models, providing evidence for their predictive power. In addition, seven out of the eight hypotheses were confirmed, particularly highlighting the positive influence of ease of use on perceived usefulness, as well as the positive relationship between intention to use and actual use of the system. The results underscore the importance of reaching a more equitable and empowering participation of women in the digital government sphere. Moreover, they provide valuable insights for formulating policies and strategies that promote the effective adoption of e-government services by the demographic group under study.

## Introduction

The advent of Information and Communication Technologies (ICTs) has revolutionized service provision in the information age. From the perspective of *public value*, electronic government—commonly known as e-government—has been implemented to optimize government structures and operations on the web [1]. Sheryazdanova et al. [2] define e-government as a new paradigm in public administration aimed at replacing the traditional bureaucratic model and providing one-stop electronic services. The objective is to make government services more accessible, mobile, and transparent.

**Funding:** The author(s) received no specific funding for this work.

Thus, ICTs have been implemented to deliver public administration services at the local, municipal, and national levels. They provide communication channels and management platforms for government agencies, businesses, and citizens, allowing them to interact individually or collectively. As a result, these online services have particularly benefited citizens [3]. Research on the e-government phenomenon has increased over the last two years with a focus on two main approaches: information systems and public administration [4]. Another relevant approach is the study of e-government adoption through the application of behavioral models and theories to understand the influencing factors and the effect of transparency on such adoption [5].

Some authors argue that the primary goal of e-government is to foster transparency and accountability in public institutions in the digital transformation age. To ensure a successful transition, it is crucial to address the various concerns that arise with these services. For example, regarding access to ICTs for certain segments of the population, numerous studies have analyzed the digital divide, which refers to the disparity in citizens' access to and skills in using these technologies [6]. Certainly, many e-government websites require improved accessibility, especially in developing countries. Therefore, efforts should be made to enable citizens of all cognitive, visual, and hearing abilities to use these e-services effectively [7].

Other studies have examined disparities in e-government adoption based on demographic factors, such as age, gender, digital literacy, and educational level [8]. However, this approach requires further exploration from the perspective of specific populations. In the context of emerging economies in Latin America, a few studies have been conducted [9]. Some of the major problems identified in these populations are related to accessibility, citizens' attention to government actions, and privacy and confidentiality—aspects that governments must consider during implementation because they affect the adoption of these technologies by citizens [7].

Research on the adoption of e-government is essential because electronic government services have demonstrated the potential to improve access to public services [7], facilitate citizen participation [10] and reduce bureaucratic barriers [11], even in vulnerable populations such as women [12]. In this sense, it is crucial to expand the existing knowledge on this area of research in order to identify factors that influence women's participation in these systems [13] and to propose solutions that promote equitable inclusion [14]. Thus, promoting women's access to e-government services can support women's empowerment in society, economic empowerment [15] and contribute to closing the existing gender gaps in the adoption of these services and the digital inclusion of women in emerging economies.

The research gap of this study lies in the lack of studies that address the adoption of e-government services in emerging economies, specifically from the perspective of young women, and this is where this study aims to contribute. This study aims to determine the factors that influence the adoption of e-government services by young women in Medellín, Colombia, where technological and social limitations are usually more evident in vulnerable populations [16]. The contribution of the study seeks to offer an analysis that enriches the existing scientific knowledge on the implementation of e-government in an emerging economy, providing specific data on women in this context. Actual use and intention to use are included simultaneously from theory to practice to obtain a comprehensive view: with intention to use revealing future motivations, while actual use allows for an assessment of the barriers and facilitators that women face when interacting with these services currently.

For this study we selected variables -perceived usefulness, perceived ease of use, attitude and subjective norm- based on the Technology Acceptance Model (TAM), which has proven to be effective in predicting technology adoption in various contexts. These variables allow capturing both the perception of e-government functionality and simplicity, as well the influence of personal attitude and social pressure. In this sense, a theoretical framework

section is presented to support the conceptual model. Subsequently, the methodology is presented, explaining how the data collection instrument was constructed and the sociodemographic information of the population is presented. The results are then presented based on a Partial Least Squares Structural Equation Model (PLS-SEM). Finally, the discussion of the analysis of the results and conclusions are presented.

## Theoretical framework

The literature emphasizes the importance of identifying and examining the factors that influence human behavior [17]. Several studies have proposed e-government adoption models based on theories and approaches widely discussed in the literature. A recent comprehensive study analyzed 25 different factors related to mobile government adoption to develop a unified model [18]. Similarly, other studies have addressed the use of open government data by citizens and the factors that impact its continuity. The findings are highly useful for policy makers and open government data providers to design or adjust user retention strategies [19].

Studies on e-government adoption have also been conducted in developed countries, such as Saudi Arabia [20]. These studies start from the assumption that e-government can fight corruption. For instance, Park and Kim [21] conducted an empirical study to determine whether e-government contributes to reducing corruption in countries. Their results revealed that it depends on the effectiveness of the legal systems. However, they observed that data-driven open government does not have a direct impact on minimizing corruption.

Developed nations such as China face a major challenge: the low interest of citizens in continuously using e-government. In this regard, Li and Shang [22] performed a thorough analysis of various factors that could shed light on citizens' intention to make continuous use of this service in the country. Other studies have focused on the multidimensional features that enable government websites to facilitate democratic processes [23]. Consequently, identifying and studying these factors is critical because they influence public acceptance and willingness to support e-government services [24].

### E-government

According to Almarabeh and AbuAli [25] e-government is defined as the use of ICTs by government authorities to provide citizens and businesses with the opportunity to interact and conduct transactions with the government. This involves the use of various electronic media, including mobile devices. The emergence of e-government is attributed to the phenomenon of globalization, the advent of ICTs, and the introduction of reforms into public administration systems. Its purpose is to provide government information and services online, allowing people to access important resources anytime, anywhere, at a fixed and low cost, and in a simpler, faster, and more convenient way [26].

The objectives of e-government have been defined within five domains: 1) Policy framework, 2) Enhanced public services, 3) High-quality and cost-effective government operations, 4) Citizen engagement in democratic processes, and 5) Administrative and institutional reform [27, 28]. By adopting e-government, governments create new public value through technological innovation. Additionally, following Santa et al. [29], it also considers accountability, transparency, interactivity, participation, and cost-effectiveness.

Meijer and Bekkers [30] for their part, define e-government as a set of techniques that aid in driving public sector modernization. In a practical sense, e-government refers to the use of ICTs to design new or redesign existing information processing and communication routes in the government. Its goal is to provide better service, improved governance in terms of electronic services to businesses and citizens, and enhanced management efficiency while

promoting democratic values and mechanisms. Thus, the transformation of digital government is framed within the scope of e-government as an indicator of innovation in the public administration of nations to provide public services to citizens [31].

**Research model and hypotheses.**   (TAM developed by Davis [32] has been used in the literature to explain the adoption of e-government among female university students. This model, which is derived from the Theory of Reasoned Action (TRA), has been proven useful in predicting user behavior when interacting with new technologies. Particularly, Mensah [17] used this model to analyze the influence of government capacity and e-government performance on the adoption of online government services. Similarly, Nofal et al. [33] proposed a framework for assessing e-government adoption considering the mediating roles of perceived usefulness and perceived ease of use in the implementation of e-government services by citizens in public sector institutions in Jordan.

Mustafa et al. [9] highlighted that TAM is one of the most commonly used approaches in e-government adoption research in developing countries. This is supported by the analysis of Ziba and Kang [34] who employed an extended version of this model in Malawi. It has also been implemented in rural contexts in Zambia to assess the adoption of e-tax systems [35]. Other examples of TAM application include its use in mobile government in Thailand [36] and e-procurement in Malaysia [37]. These diverse applications of TAM provide a stimulus for researchers to continue relying on and using the model as a foundation for explaining user adoption of new technologies [17]. Therefore, this study proposes the research model depicted in Fig 1.

*Perceived usefulness*. Perceived usefulness has been shown to positively affects users' behavioral intention [38]. It has been defined as the "subjective probability that user will increase its productivity using a specific application in its work" [39]. In other words, users believe that employing e-government services can help them perform their job more efficiently and effectively [26]. Perceived usefulness is often associated with easier work, increased productivity, improved efficiency, and higher performance [40]. In this regard, TAM studies have demonstrated that perceived usefulness has a direct impact on behavioral intention [41] and a positive effect on people's attitude towards technology [40].

The concept of perceived usefulness is related to the ability to gain an advantage, i.e., something that is useful. In the context of technology, users perceive a technological solution as useful if they believe that its use can improve their job performance [42] and a positive effect on

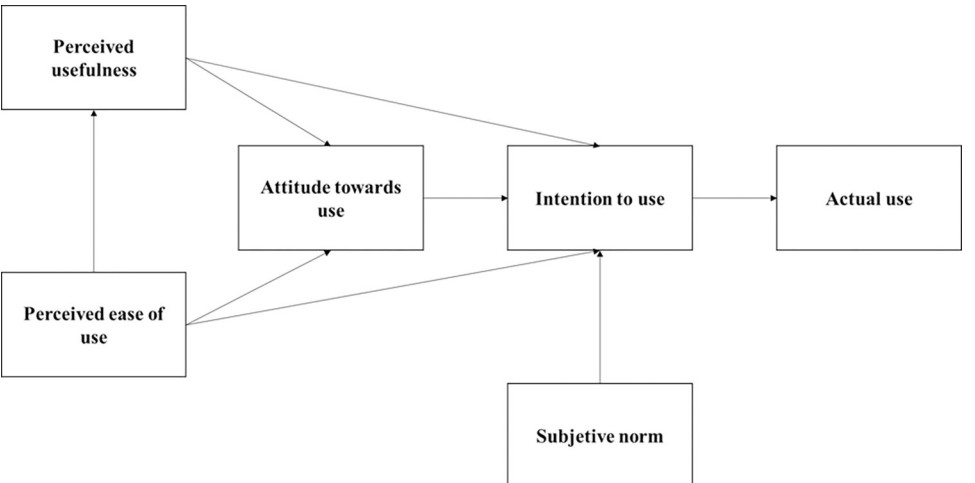

**Fig 1. Model of e-government adoption by female university students.**

people's attitude towards technology [43]. In addition, its influence on the adoption of e-government and meta-government has been demonstrated. According to Al-Adwan [44], if citizens perceive that the meta-government system offers valuable services, efficient processes, and better access to government resources, they will tend to consider it useful and beneficial. Therefore, the following research hypotheses are formulated:

H1. Perceived usefulness has a positive influence on female university students' attitude towards the use of e-government services

H2. Perceived usefulness has a positive influence on female university students' intention to use e-government services

*Perceived ease of use.* Perceived ease of use is the extent to which individuals believe that using a technological tool is 'free of effort' [45]. Consequently, it has a direct impact on their intention to adopt an information system. Similarly, Gunawan et al. [46] define this concept as the level at which an individual is certain that using a system does not require any effort. In the context of e-government, perceived ease of use refers to the simplicity of learning how to use electronic services, offering straightforward operations [47].

Previous studies have shown that perceived ease of use has a positive impact on the adoption of e-government services, as well as on users' attitude toward behavior and perceived usefulness [48]. This means that the more users perceive a system to be easy to use, the greater their interest in using it. Al-Adwan [44] found that e-government is perceived as more useful when accompanied by ease of use. Furthermore, when users perceive e-government systems as easy to use, they also perceive them as more useful. In other words, if users perceive that a system is easy to operate, their attitude towards its use will be more positive [49]. In this context, previous studies have also found that perceptions of ease of use and usefulness are of paramount importance in fostering the intention to adopt meta-governance [50].

As explained by Warsono et al. [51] in the context of e-government, the level of convenience and practicality of these services is a determining factor in people's behavior when using them. In this sense, if the e-government service model is easy to use, it will increase the effectiveness, efficiency, and convenience of the community in accessing public services. Therefore, previous studies have found that the perceived ease of use of an electronic service, such as e-government, is a practical quality that significantly influences the attitude of users [52]. Accordingly, the following research hypotheses are proposed:

H3. Perceived ease of use has a positive influence on female university students' perceived usefulness of e-government services

H4. Perceived ease of use has a positive influence on female university students' attitude towards the use of e-government services

H5. Perceived ease of use has a positive influence on female university students' intention to use e-government services

*Attitude towards use.* In a broader context, the term 'attitude toward behavior' refers to an individual's positive or negative evaluation of performing a specific behavior [53]. This concept is critical to understanding how individuals perceive and respond to technologies, particularly in the TAM domain. The influence of individuals' attitude on behavioral intention has been a relevant topic in TAM-based research. Within the framework of this research, attitude towards the adoption of e-government systems refers to the level at which users (in this case, women) express a positive or negative assessment regarding their participation or interaction with these systems [54].

Previous studies have demonstrated that users' attitude has a direct and significant impact on their intention to adopt a system or technology [55]. Specifically, users' attitude towards e-government services has been found to predict their usage intentions [56]. This approach is supported by recent studies that have shown that TAM factors remain useful in predicting citizens' intentions to use e-government services during the COVID-19 pandemic. In particular, attitudes toward e-government, according to Nguyen [57], play a mediating role in the relationship between perceived usefulness, perceived risk, and citizens' intentions to use these services. Based on this information, the following hypothesis is formulated:

H6. Attitude towards use has a positive influence on female university students' intention to use e-government services

*Subjetive norm*. Following AL-Nawafleh et al. [58] subjective norm refers to the likelihood of an individual engaging in a certain behavior, such as adopting e-government, based on the opinion of other individuals who are important to them (e.g., family members, friends, colleagues). This means that the perception of others exerts a motivating influence on their intention to use a technology, as noted by Kumar et al. [59]. In this context, Kumar, Mukherjee et al. [60] define this factor as "the degree to which an individual believes that others think he or she should use e-government services." This normative pressure comes from the family or friends of the associated members, who influence the intention to use e-government.

The study by Iong and Phillips [61] revealed that a crucial factor influencing e-government service adoption behavior is external influence, i.e., subjective norm, particularly that exerted by superiors in the workplace and government policies. Studies on e-government adoption have shown that subjective norm significantly and positively impacts the intention to use e-government services, as supported by Zahid and Haji Din [62]. In consequence, the following hypothesis is formulated:

H7. Subjective norm has a positive influence on female university students' intention to use e-government services

*Intention to use*. As defined in the study by Mailizar et al. [63] behavioral intention is "a cognitive process of individuals' readiness to perform a specific behaviour and is an immediate antecedent of usage behaviour." In addition, this notion is associated with individuals' tendency to accept and acknowledge technology, which becomes a relevant indicator of continued use intention, as explained by Abu-Taieh et al. [64]. In a recent study, Hooda et al. [65] analyzed the factors that influence the intention to use e-government services and their correlation with e-government system usage behavior. The findings led to the following research hypothesis:

H8. Intention to use has a positive influence on female university students' actual use of e-government system

## Methodology

To achieve the objective of this study, we employed a quantitative methodology with a correlational approach. We used a TAM-based structural equation model to investigate the adoption of e-government services by female university students. For this purpose, we administered a questionnaire to a sample of 223 women from three higher education institutions in Medellín, Colombia: Instituto Tecnológico Metropolitano (63%), Institución Universitaria Pascual Bravo (17%), and Institución Universitaria Colegio Mayor de Antioquia (20%). These three educational institutions are affiliated with the Medellín Mayor's Office. The study sampling

was non-probabilistic, given the need to access the target population in an efficient manner, given the time and resources available for the project.

Specifically, 70% of the respondents were enrolled in the first four semesters of their academic programs. In addition, 57% of the participants were employed and 14% were self-employed while also studying. The remaining 29% were solely dedicated to their studies. In terms of socioeconomic status, 13% of the respondents belonged to stratum 1 (low-low), 44% to stratum 2 (low), 33% to stratum 3 (medium-low), 7% to stratum 4 (medium), and 2% to stratum 5 (medium-high). With regard to the age of the participants, 69% were between 16 and 27 years old, 28% between 28 and 40 years old, and 3% between 41 and 57 years old.

We structured the questionnaire based on the TAM model and then adapted it to the research context. Participants rated a series of statements on a five-point Likert scale ranging from 'Strongly disagree' to 'Strongly agree.' To ensure clarity, we conducted a pilot test with fifteen students and made the necessary corrections before administering the final questionnaire.

To gather information, we used a questionnaire that also detailed the objective of the study. Data collection was done through an online survey on Google Forms. The survey contained an informed consent form and data collection was conducted in 2022 and 2023. Before implementing the questionnaire, experts were consulted to assess the relevance of the research questions. Their recommendations were then incorporated, and a pilot test involving 15 participants was conducted to evaluate how well the target population understood the questionnaire. After incorporating the suggestions, the survey was administered. Participants were informed that the research aimed to identify factors influencing the adoption of e-government. Anonymity was emphasized, and participants were assured that no compensation was required or provided for completing the survey. Finally, they were told that there were no right or wrong answers in the questionnaire. The data collection instrument used can be seen in Table 1. This study and informed consent were reviewed and approved by the Ethics Committee of the Institución Universitaria Escolme (Ethics Committee of Institución Universitaria ESCOLME,) with the ACTA 01 date 10042023, before the study began. Informed consent was given orally, explaining the aim of the research and all the ethical aspects involved.

## Results

This study used the statistical software SmartPLS 4 to perform Partial Least Squares Structural Equation Modeling (PLS-SEM). The purpose of this analysis was to assess the reliability and validity of the measurement model and to test the hypotheses of the structural model. We run factor analyses to assess both convergent and discriminant validity and then tested the hypotheses simultaneously [66]. The PLS-SEM approach was selected because it is variance-based and is more flexible and suitable for theoretical exploration compared to covariance-based structural equation methods (CB-SEM), which require normal distributions and larger sample sizes. In addition, because of its focus on model prediction.

The Common Factor Method was used to mitigate the potential bias of the common method. During the analysis phase, we examined the convergent and discriminant validity, as well as the reliability of the measurement model. Moreover, when evaluating the structural model, we analyzed its hypotheses and predictive ability. This comprehensive approach provides a thorough understanding of the robustness and effectiveness of the proposed model.

### Measurement model

In the first stage, we measured factor loadings as an indicator of convergent validity to ensure appropriate measurement of the indicators in relation to their respective latent constructs. The

**Table 1. Factors and indicators for data collection.**

| Factor | Item | Description |
|---|---|---|
| Perceived Ease-of-Use | FUP1 | I am facilitated in the use of digital tools |
| | FUP2 | I have access to electronic devices with Internet |
| | FUP3 | It is very easy to use the tools and services in the government pages. |
| | FUP4 | I have easy access to the Internet and devices (PC, cell phone, Tablet) to make consultations and procedures through e-government |
| | FUP5 | Normally, I do everything through the Internet |
| | FUP6 | The interface of the government pages are user friendly |
| Perceived Usefulness | PU1 | E-government sites are reliable and secure |
| | PU2 | The government websites offer me different options to answer my concerns |
| | PU3 | I have found very useful tools and services on the government websites. |
| Subjetive Norm | SN1 | I found the government sites on the recommendation of a family member, friend or acquaintance. |
| | SN2 | I saw that someone else used online government services and I did too. |
| Attitude Towards Use | AHU1 | E-government services allow for agility, speed and reliability of procedures. |
| | AHU2 | Using e-government tools, I do not need to travel to carry out procedures or request services. |
| | AHU3 | Using e-government tools, I can carry out the procedure or access services without having to travel or wait. |
| | AHU4 | Using e-government tools, I do not need to pay for tickets or spend fuel to carry out procedures. |
| | AHU5 | E-government information, services and procedures are of good quality. |
| Intention to Use | IU1 | I am interested in the information I find on government websites. |
| | IU2 | I have needed to perform a procedure or request services on government websites. |
| | IU3 | I found the government sites using a search engine. |
| | IU4 | I use e-government tools to avoid going to government offices and prevent getting diseases (Covid-19). |
| Actual Use of the System | UR1 | In the future, government procedures and services will be 100% virtual. |
| | UR2 | E-government is a valuable tool in times of the Covid-19 pandemic. |
| | UR3 | I intend to do business on government websites in the short term. |
| | UR4 | I would like government information, procedures and services to be available only virtually (zero face-to-face). |

Source: own elaboration based on Davis [32]

results of these measurements are shown in Table 2, which contains the cross-loadings. It is important to note that these loadings always fall within the range of -1 to 1.

Based on Amora [67] explanation of reflective latent constructs, high factor loadings and low cross-loadings were expected. The literature generally establishes a fundamental criterion for acceptable convergent validity of a measurement model, often requiring loadings to reach or exceed 0.5. The results demonstrate that the criterion was met exceptionally well, as all factor loadings exceeded the 0.7 threshold. This supports the proposed measurement model's robustness and convergent validity.

Subsequently, in the evaluation of convergent validity, we applied the Average Variance Extracted (AVE) and Variance Inflation Factor (VIF) statistics. It is suggested that factor loadings should be greater than 0.7 for the AVE, as the square of this value indicates that at least 50% of the variable's variance is included in the construct score. Specialized researchers support an AVE value higher than 0.5, indicating that more than half of the indicator's variance is included in the latent construct score [68]. Additionally, the VIF is used to assess the possible

**Table 2. Cross-loadings.**

|  | Perceived usefulness | Perceived ease of use | Attitude towards use | Subjective norm | Intention to use | Actual use |
|---|---|---|---|---|---|---|
| PU1 | **0.891** | 0.721 | 0.683 | 0.405 | 0.666 | 0.608 |
| PU2 | **0.927** | 0.710 | 0.709 | 0.436 | 0.598 | 0.577 |
| PU3 | **0.882** | 0.697 | 0.690 | 0.484 | 0.641 | 0.615 |
| PEU1 | 0.473 | **0.789** | 0.528 | 0.273 | 0.597 | 0.590 |
| PEU2 | 0.451 | **0.780** | 0.462 | 0.226 | 0.460 | 0.450 |
| PEU3 | 0.709 | **0.795** | 0.689 | 0.501 | 0.722 | 0.619 |
| PEU4 | 0.540 | **0.821** | 0.582 | 0.314 | 0.540 | 0.537 |
| PEU5 | 0.521 | **0.702** | 0.526 | 0.227 | 0.469 | 0.503 |
| PEU6 | 0.836 | **0.753** | 0.685 | 0.470 | 0.606 | 0.611 |
| ATU1 | 0.728 | 0.679 | **0.860** | 0.432 | 0.660 | 0.554 |
| ATU2 | 0.664 | 0.659 | **0.918** | 0.401 | 0.633 | 0.562 |
| ATU3 | 0.689 | 0.685 | **0.908** | 0.426 | 0.652 | 0.564 |
| ATU4 | 0.630 | 0.641 | **0.866** | 0.399 | 0.646 | 0.626 |
| ATU5 | 0.584 | 0.610 | **0.720** | 0.521 | 0.578 | 0.649 |
| SN1 | 0.464 | 0.437 | 0.464 | **0.925** | 0.567 | 0.440 |
| SN2 | 0.444 | 0.407 | 0.472 | **0.924** | 0.564 | 0.437 |
| IU1 | 0.701 | 0.708 | 0.713 | 0.507 | **0.858** | 0.623 |
| IU2 | 0.465 | 0.531 | 0.555 | 0.540 | **0.813** | 0.610 |
| IU3 | 0.396 | 0.547 | 0.459 | 0.456 | **0.787** | 0.503 |
| IU4 | 0.721 | 0.659 | 0.684 | 0.512 | **0.838** | 0.587 |
| AU1 | 0.550 | 0.662 | 0.516 | 0.331 | 0.541 | **0.782** |
| AU2 | 0.573 | 0.634 | 0.648 | 0.326 | 0.629 | **0.801** |
| AU3 | 0.505 | 0.529 | 0.551 | 0.395 | 0.567 | **0.847** |
| AU4 | 0.453 | 0.424 | 0.412 | 0.459 | 0.462 | **0.698** |

Source: Own work using SmartPLS 4.

presence of multicollinearity problems in the model. It is established that the value of each indicator should not exceed a threshold of 5 [69] to ensure the model's robustness. These criteria provide a rigorous and comprehensive evaluation of the model's convergent validity, ensuring the reliability of the results.

Furthermore, to assess internal consistency, we employed Cronbach's alpha (CA) and Composite Reliability (CR). These two measures have a similar interpretation, indicating that the latent constructs should attain values equal to or greater than 0.7, which are considered acceptable, and values of 0.8, which are considered very satisfactory [70]. The results shown in Table 3 demonstrate that each of these criteria is rigorously met, thereby consolidating the convergent validity of the proposed model.

We measured the model's discriminant validity using the Heterotrait-Monotrait (HTMT) index, as suggested by Hair and Alamer [71]. This index determines the extent to which a construct is conceptually distinct from other constructs in the study. Each theoretical construct in the model is unidimensional, measuring a singular concept and showing only a small overlap in variances. Preferably, the HTMT ratio of the correlations should be < 0.85 or < 0.90. Table 4 shows that this criterion is satisfactorily met, confirming the discriminant validity of the proposed model. This finding supports the idea that each theoretical construct is distinct and measures a unique concept within the study framework.

**Table 3. Factor loadings, AVE, and model's reliability.**

| Factor | Indicator | FL | VIF | CA | CR | AVE |
|---|---|---|---|---|---|---|
| Perceived usefulness | UP1 | 0.891 | 2.449 | 0.883 | 0.883 | 0.810 |
| | UP2 | 0.927 | 3.232 | | | |
| | UP3 | 0.882 | 2.299 | | | |
| Perceived ease of use | FUP1 | 0.789 | 2.358 | 0.868 | 0.877 | 0.600 |
| | FUP2 | 0.780 | 2.832 | | | |
| | FUP3 | 0.795 | 1.992 | | | |
| | FUP4 | 0.821 | 2.681 | | | |
| | FUP5 | 0.702 | 1.573 | | | |
| | FUP6 | 0.753 | 1.794 | | | |
| Attitude towards use | AHU1 | 0.860 | 2.828 | 0.908 | 0.932 | 0.735 |
| | AHU2 | 0.918 | 4.556 | | | |
| | AHU3 | 0.908 | 4.051 | | | |
| | AHU4 | 0.866 | 2.777 | | | |
| | AHU5 | 0.720 | 1.614 | | | |
| Subjective norm | NS1 | 0.925 | 2.007 | 0.829 | 0.829 | 0.854 |
| | NS2 | 0.924 | 2.007 | | | |
| Intention to use | IU1 | 0.858 | 2.138 | 0.843 | 0.851 | 0.680 |
| | IU2 | 0.813 | 1.888 | | | |
| | IU3 | 0.787 | 1.813 | | | |
| | IU4 | 0.838 | 2.026 | | | |
| Actual use | UR1 | 0.782 | 1.570 | 0.790 | 0.800 | 0.615 |
| | UR2 | 0.801 | 1.684 | | | |
| | UR3 | 0.847 | 2.004 | | | |
| | UR4 | 0.698 | 1.484 | | | |

Source: Own work using SmartPLS 4. Note: FL > 0.7; VIF < 5; CA > 0.7; CR > 0.7; AVE > 0.5.

## Structure model

During this phase, we tested the hypotheses and assessed the model's predictive capacity. This procedure, performed by Bootstrapping, allowed us to examine the relationships between the key factors that influence the acceptance of digital technologies in e-government services targeted at female university students. The results of this analysis include the path coefficient, which describes the strength of the relationship between the constructs, as well as the p-value and T-value significance test, as suggested by Purwanto and Sudargini [70].

**Table 4. Discriminant validity.**

| | Actual use | Attitude towards use | Intention to use | Perceived ease of use | Perceived usefulness | Subjective norm |
|---|---|---|---|---|---|---|
| Actual use | | | | | | |
| Attitude towards use | 0.806 | | | | | |
| Intention to use | 0.855 | 0.836 | | | | |
| Perceived ease of use | 0.850 | 0.841 | 0.846 | | | |
| Perceived usefulness | 0.795 | 0.861 | 0.802 | 0.866 | | |
| Subjective norm | 0.596 | 0.588 | 0.731 | 0.509 | 0.574 | |

Source: Own work using SmartPLS 4. Note: HTMT < 0.85.

**Table 5. Hypothesis test.**

| | Hypothesis | Path β-value | T-value | p-value | Acceptance |
|---|---|---|---|---|---|
| H1 | Perceived Usefulness → Attitude Towards Use | 0.444 | 5.452 | 0.000 | Yes |
| H2 | Perceived Usefulness → Intention to Use | 0.097 | 1.358 | 0.175 | No |
| H3 | Perceived Ease-of-Use → Perceived Usefulness | 0.788 | 31.413 | 0.000 | Yes |
| H4 | Perceived Ease-of-Use → Attitude Towards Use | 0.416 | 5.542 | 0.000 | Yes |
| H5 | Perceived Ease-of-Use → Intention to Use | 0.342 | 4.280 | 0.000 | Yes |
| H6 | Attitude Towards Use → Intention to Use | 0.266 | 3.589 | 0.000 | Yes |
| H7 | Subjetive Norm → Intention to Use | 0.273 | 4.599 | 0.000 | Yes |
| H8 | Intention to Use → Actual Use of the System | 0.707 | 16.289 | 0.000 | Yes |

Source: own elaboration based on SmartPLS 4. Note: T value > 1.96; p value < 0.05.

In this context, we evaluated the statistics using β-values (> 0.005), T-value (> 1.96), and p-value (< 0.05), according to the criteria established by Kock [72]. The results of these evaluations are detailed in Table 5 and Fig 2, providing a clear picture of the statistical significance and direction of the relationships between the latent constructs in the proposed model.

The analysis of this model provides insight into its predictive capacity through the coefficient of determination $R^2$. This estimate reflects the variance in the results, which is explained by the predictor constructs. It is important to interpret the values of $R^2$ according to their results. In this sense, values between 0 and 0.10, 0.11 and 0.30, 0.30 and 0.50, and > 0.50 indicate weak, modest, moderate, and strong explanatory power, respectively. Fig 2 depicts the results for the endogenous latent constructs, demonstrating their strong predictive value for Intention to use, Perceived usefulness, Attitude towards use, and Actual use. This finding strongly confirms the predictive power of the proposed model, supporting the conclusions of previous studies by Purwanto and Sudargini [70] and Hair and Alamer [71].

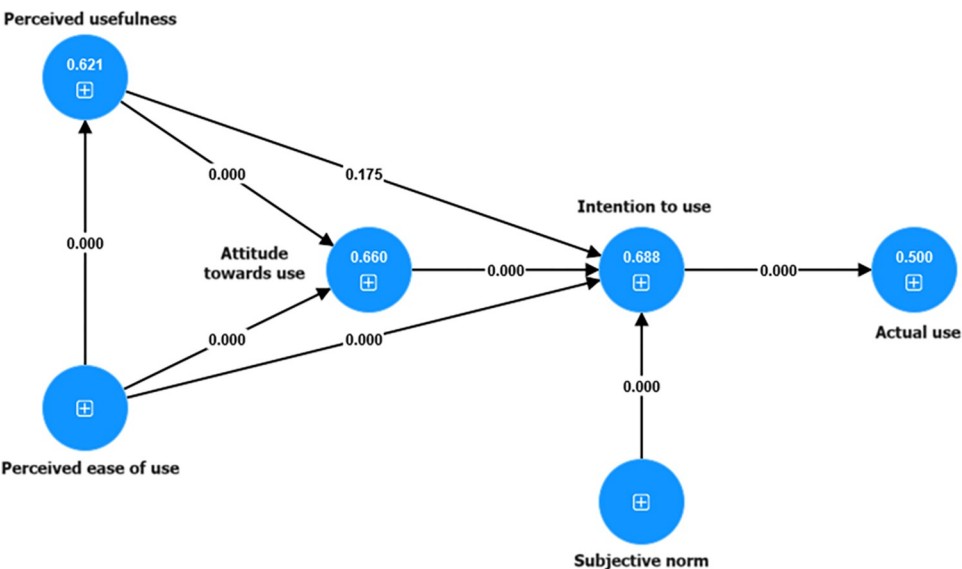

**Fig 2. Statistical hypothesis test and predictive value of the model of e-government adoption by female university students.**

We used the $Q^2$ statistic to measure the robustness of the model's predictive power. High $Q^2$ values indicate appropriateness of exogenous latent variables in predicting endogenous variables [68]. However, researchers have provided specific guidelines for values indicating weak, moderate, and strong levels of prediction: 0.02, 0.15, and 0.35, respectively [68]. This study obtained significant values for Intention to use (0.637), Perceived usefulness (0.616), Attitude towards use (0.579), and Actual use (0.508). These results further confirm the outstanding model's predictive ability, thus supporting its ability to estimate the future use of e-government services by female university students.

In relation to the hypotheses of the model, seven out of the eight hypotheses were satisfactorily supported. We identified a highly significant relationship between Perceived ease of use and Perceived usefulness (H3), as well as between Intention to use and Actual use (H8). Regarding attitude, we observed a significant positive relationship with both Perceived usefulness (H1) and Perceived ease of use (H4). However, it is worth highlighting that Perceived usefulness did not prove to be a determining factor in the intention to use e-government services (H2). In contrast, Subjective norm (H7), Perceived ease of use (H5), and Attitude towards use (H6) proved to be influential factors in the intention to use. These results underscore the importance of the latter constructs in the adoption of e-government by female university students.

## Discussion

The literature on e-government implementation suggests that, in general terms, women have lower adoption rates of digital government platforms. However, recent studies, such as the one conducted by Soni and Mitchell [73], have shown that women actively participate in direct interaction with government websites. This participation helps to bridge the digital gender gap and increases both perceived usefulness and perceived ease of use, which, in turn, promotes the acceptance and use of digital government services among women living in rural areas. The above is consistent with the results of this study, which highlights the importance of perceived ease of use (H5) and attitude towards use (H6) to the process of e-government adoption by women.

Different studies support the idea that women actively use government websites and official applications, indicating a positive acceptance of e-government [8]. In emerging economy contexts, women are typically underrepresented in citizen participation compared to developed countries. However, previous studies have also suggested that gender may positively moderate the relationship between intention to use and actual adoption (H8) of e-government services [74]. Moreover, this study, which focused on an emerging economy, highlights the active participation of women in e-government services. This finding suggests a remarkable reduction in the global gender gap, especially in terms of women's engagement with digital government in developing economies.

Studies such as the one conducted by Xie et al. [75] support the hypothesized relationships proposed in TAM, including the influence of perceived ease of use on perceived usefulness (H3). This means that accessible and user-friendly government platforms and processes are perceived as more useful by women. Furthermore, the relationship between intention to use and actual use (H8) is also noteworthy because it demonstrates that women's prior attitudes, perceptions, and motivations have an impact on the adoption and use of these platforms. These findings are in line with those reported by ELKheshin and Saleeb [76].

This study found that perceived usefulness is not a determining factor for women's use of e-government (H2). These findings could indicate that, in this specific context, women do not consider the practical or functional benefits of e-government, thus, although they may

recognize that the use of e-government services is useful, this recognition does not directly translate into a greater intention to use them. These findings run counter to previous studies that have highlighted the critical role of perceived usefulness in the adoption of these technologies [77]. In fact, perceived usefulness has not only played a full mediating role but is expected to be the most important ex post factor influencing user intention to continue using the e-government service [78]. Therefore, it is a factor to be strengthened in the context of the city of Medellin.

Previous studies support the hypothesis that perceived usefulness has a positive influence on attitude (H1) of women by proving that attitude is a powerful mediator between beliefs (perceived usefulness and perceived ease of use) and intention to use e-government services [79]. Thus, this finding indicates that, if women recognize that e-government can improve their access to public services and simplify processes, they tend to have a more positive disposition towards the idea of using these systems. This is, in short, a positive perception. Attitude has also been shown to be influenced by perceived ease of use (H4). Thus, if users perceive that interacting with the platform does not require considerable effort or advanced technological skills, they develop a more favorable and positive attitude towards the use of e-government, which is in line with the findings of Oztaskin et al. [80] indicating that age, gender and education positively impact the perceived ease of use of e-government services.

Finally, women's intention to use e-government was influenced by the subjective norm (H8). This indicates that if women feel that people important to them (such as family, friends, or colleagues) support or value the use of e-government, they will be more inclined to use it. These findings suggest that social influence or recommendations from their environment significantly influence intention to use, as this factor has also been shown to be relevant in other studies. The research by Batara et al. [81] found that social influence positively affects the intention to adopt process redesign, organizational structuring, and cultural and behavioral change in city government in the face of e-government transformation. Therefore, this finding is also supported by the literature.

## Theoretical implications

In recent years, there has been a significant increase in e-government studies, mainly driven by the various benefits that these types of services provide globally. However, unlike the majority of research works that address e-government adoption in general, this study focused on female university students, who represent the future users of these services. In addition, it is important to note that this study offers a distinctive point of view from an emerging economy in Latin America, where no research of this type had been conducted on this specific population.

In this context, TAM makes it possible to analyze women's behavior towards this phenomenon. The results demonstrate that the model is relevant and provides an accurate explanation of the behavior of female university students in Colombia regarding the actual use of e-government. From a theoretical perspective, this study contributes to the existing knowledge on the adoption of e-government services by women—an underrepresented population group in many emerging economies, as evidenced in previous analyses [73]. In this sense, the findings not only enhance the understanding of the psychological factors that influence the adoption process, but also offer valuable insights for bridging gaps in access to these services.

## Practical implications

Likewise, the results of this study provide useful feedback for the design of effective strategies for promoting e-government services, specifically among female university students in

Colombia. In addition, they help identify perception barriers, as in the case of perceived usefulness, and demonstrate the importance of user-friendliness and accessibility. They are also helpful in understanding the relationship between intention and actual use, which is crucial for the creation of more inclusive policies and programs.

These findings underscore the need to address the perceptions of usefulness and ease of use to enhance female university students' acceptance of and active participation in e-government. Understanding the dynamics between intention and actual use can lead to the establishment of more effective policies and programs, thus contributing to gender equality in accessing and leveraging information technologies. This, ultimately, can foster more equitable and empowering involvement of women in the digital government arena [82].

## Conclusions

The implementation of e-government in the digital age has led to a more accessible, mobile, and transparent paradigm in public administration. Although these online services offer major benefits, recent studies have focused on addressing specific challenges, such as the digital divide and the lack of accessibility in emerging economies. This study explored the factors that influence young women's adoption of e-government services in Medellín, Colombia. As a result, it emphasized the importance of understanding the dynamics of specific populations in developing economies to improve the implementation of and equitable access to these services. The findings contribute significantly to scientific knowledge on e-government adoption from a female perspective in emerging economy contexts.

To achieve the objective of the study, a questionnaire was administered to 223 female university students from three higher education institutions in Medellín, following the TAM model. The results validated the model through convergent and discriminant analyses, highlighting its robust reliability. In addition, seven out of the eight hypotheses were confirmed. Notably, the most significant relationships were identified between perceived ease of use and perceived usefulness, as well as between intention to use and actual use of e-government services.

With respect to the factors that influence the intention to use e-government services, the results emphasized the importance of Perceived ease of use, Subjective norm, and Attitude towards use. It is worth mentioning that a negative relationship was observed with Perceived usefulness. These results enhance the understanding of the key factors that impact the adoption of e-government services by female university students, thereby contributing to the existing literature in this field.

### Limitations

Regarding limitations, it is important to note that this study employed the TAM model and incorporated Subjective norm as an external variable, which was found to have a positive influence on the intention to use e-government. However, to propose a unified model of e-government adoption and gain a more complete understanding of this phenomenon, it was necessary to explore it from other theoretical approaches, such as the Theory of Planned Behavior (TPB), the Theory of Reasoned Action (TRA), and the Unified Theory of Acceptance and Use of Technology (UTAUT) [40, 17–76].

Future studies could expand the research scope to other geographies, as this study focused on one Colombian city, which limits the generalizability of the results. Exploring possible variations in e-government adoption among regions of the country would provide a more comprehensive and representative picture of the phenomenon in the Colombian context.

## Author Contributions

**Conceptualization:** Paula Andrea Rodríguez-Correa, Alejandro Valencia-Arias, Jorge Tomás Cumpa Vásquez, Toño Eldrin Alvites Adan.

**Data curation:** Paula Andrea Rodríguez-Correa.

**Formal analysis:** Carlos Alberto Méndez-Rivera, Alejandro Valencia-Arias.

**Funding acquisition:** Orfa Nidia Patiño-Toro.

**Investigation:** Alejandro Valencia-Arias.

**Methodology:** Orfa Nidia Patiño-Toro, Alejandro Valencia-Arias, Ada Lucia Gallegos Ruiz Conejo.

**Project administration:** Orfa Nidia Patiño-Toro.

**Resources:** Alejandro Valencia-Arias, Aarón Oré León, Toño Eldrin Alvites Adan.

**Software:** Ada Lucia Gallegos Ruiz Conejo, Jorge Tomás Cumpa Vásquez.

**Supervision:** Carlos Alberto Méndez-Rivera, Jorge Tomás Cumpa Vásquez, Toño Eldrin Alvites Adan.

**Validation:** Aarón Oré León, Toño Eldrin Alvites Adan.

**Writing – original draft:** Paula Andrea Rodríguez-Correa, Carlos Alberto Méndez-Rivera, Orfa Nidia Patiño-Toro, Alejandro Valencia-Arias, Ada Lucia Gallegos Ruiz Conejo, Aarón Oré León, Jorge Tomás Cumpa Vásquez, Toño Eldrin Alvites Adan.

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
