## [Decision Letter · Decision Letter 0]

9 Sep 2024

PONE-D-24-32042Factors influencing the adoption of e-government by female university studentsPLOS ONE

Dear Dr. Valencia-Arias,

Thank you for submitting your manuscript to PLOS ONE. After careful consideration, we feel that it has merit but does not fully meet PLOS ONE’s publication criteria as it currently stands. Therefore, we invite you to submit a revised version of the manuscript that addresses the points raised during the review process.

We note that one or more reviewers has recommended that you cite specific previously published works. As always, we recommend that you please review and evaluate the requested works to determine whether they are relevant and should be cited. It is not a requirement to cite these works. We appreciate your attention to this request.

We look forward to receiving your revised manuscript.

Kind regards,

Mingming Li

Academic Editor

PLOS ONE

2. For studies reporting research involving human participants, PLOS ONE requires authors to confirm that this specific study was reviewed and approved by an institutional review board (ethics committee) before the study began. Please provide the specific name of the ethics committee/IRB that approved your study, or explain why you did not seek approval in this case.

Reviewers' comments:

Reviewer's Responses to Questions

**Comments to the Author**

1. Is the manuscript technically sound, and do the data support the conclusions?

Reviewer #1: Yes

Reviewer #2: Partly

2. Has the statistical analysis been performed appropriately and rigorously? 

Reviewer #1: Yes

Reviewer #2: Yes

3. Have the authors made all data underlying the findings in their manuscript fully available?

Reviewer #1: Yes

Reviewer #2: Yes

4. Is the manuscript presented in an intelligible fashion and written in standard English?

Reviewer #1: Yes

Reviewer #2: Yes

5. Review Comments to the Author

Reviewer #1: Thank you for submitting your paper to PLOS ONE. This is a well-written paper that covers an interesting and timely topic. However, to ensure the quality and impact of this work, a set of corrections must be made before it can be considered for publication:

Research Problematization and Contributions: The research problem and contributions need to be presented in a clearer and more concise manner. Clearly articulating the gap in the literature that your study addresses and the unique contributions your research makes will enhance the paper's overall impact.

Hypotheses Development: The proposed hypotheses should be strengthened and must be theoretically supported. Consider providing a more robust rationale for each hypothesis, drawing on relevant theoretical frameworks and empirical evidence.

Literature Review and Theoretical Foundation: The literature review can be significantly strengthened by incorporating recent and relevant literature. This includes, but is not limited to, the following sources:

"Navigating the Roadmap to Meta-Governance Adoption" (DOI: 10.1108/GKMC-02-2024-0105)

"The Government Metaverse: Charting the Coordinates of Citizen Acceptance" (DOI: 10.1016/j.tele.2024.102109)

Retesting the TAM Model: The paper attempts to retest the Technology Acceptance Model (TAM), which has been extensively tested in numerous studies. As per the recommendations of Venkatesh, the retesting of TAM and UTAUT in marginally different contexts does not add substantial value to the ongoing discussion on technology adoption. It is recommended that you either justify the retesting in this context more convincingly or consider exploring a different or modified model that offers novel insights.

Discussion Section: The paper must include a detailed discussion section that elaborates on the main findings. This should be followed by a thorough exploration of both the theoretical and practical implications of your research. Doing so will help to contextualize your results within the broader field and provide clear guidance for future research and practice.

Conclusion and Research Limitations: The paper should conclude with a strong conclusion that summarizes the key findings. Additionally, it is important to address the limitations of your study and propose directions for future research. This will not only demonstrate the rigor of your work but also guide others in the field.

Reviewer #2: 1. It would be good to include the recent development of e-government by including the citations from 2023 to 2024. Why it is important to conduct a study of e-government? What is the research gap, motivation of study and contribution of this study? Why actual use and intention to use are included concurrently in this study?

2. The determinants are perceived usefulness, perceived ease of use, attitude and subjective norm. In introduction, include why these variables are selected and what are the differences of this study compared to other study? Why perceived behavioral control is not included in this study?

3. For research model and hypotheses, include the recent review of the literature such as in year 2023 and 2024 for each of the variable/construct. Further discuss the underpinning theories of the research framework. For H7, write more about the perceived connectedness and its relation to perceived ease of use.

4. H7. Perceived connectedness of smart home services has a positive effect on the perceived ease of use of the services. Suppose this hypothesis is H6? The proposed research framework in Figure 2 does not show this hypothesis. Please confirm if this hypothesis is tested in this study.

5. There are two H7. Please confirm if H7. Subjective norm has a positive influence on female university students’ intention to use e-government services and H8. Intention to use has a positive influence on female university students’ actual use of e-government system are supposed to be H7 and H8? Align the findings in Table 4. Indicate H1 to H8 in Table 4.

6. There is a lack of description of the data collection process, for instance, the use of questionnaire? Online? Face to face?

7. What are the sampling techniques and how to determine the minimum sample size?

8. It is recommended that to include the source of instruments/ measurement items in a table or describe it for each of the constructs.

9. What are the procedures used to analyze the data? Further discuss why PLS-SEM is selected as the estimation procedure. Did common method bias check is performed?

10. For structural model results presentation, include the interpretation of each of the hypothesis. Indicate which hypothesis is supported and not supported. Align Table 4 with Figure 2 as well as the hypotheses proposed in the section of ‘research model and hypotheses’.

11. For discussion, include the justification for each of the hypothesis. For instance, H1 Perceived usefulness has a positive influence on female university students’ attitude towards the use of e-government services. Indicate how does perceived usefulness is related to attitude among female, supported by which previous study. Same applies to H2 to H8. Kindly cross check and align the hypothesis across whole manuscript.

12. What are the theoretical and practical implications?

13. Include limitations and recommendation for future study after the conclusion.

6. PLOS authors have the option to publish the peer review history of their article (what does this mean?). If published, this will include your full peer review and any attached files.

Reviewer #1: No

Reviewer #2: No

---

## [Author Response · Author response to Decision Letter 0]

12 Dec 2024

Please see the attached file "Response to reviewers" with the timely response for each comment made by the reviewers

---

## [Decision Letter · Decision Letter 1]

3 Jan 2025

影响女大学生采用电子政务的因素

PONE-D-24-32042R1

尊敬的 Valencia-Arias 博士

我们很高兴地通知您，您的手稿已被科学地判断为适合出版，并且一旦满足所有未完成的技术要求，将被正式接受出版。

一周内，您将收到一封电子邮件，详细说明所需修改内容。修改完成后，您将收到正式的录用通知，您的稿件将安排出版。

Kind regards,

Mingming Li

Academic Editor

PLOS ONE

Additional Editor Comments (optional):

Please check the further grammar and several mistakes when you proofread in the future.

Reviewers' comments:

Reviewer's Responses to Questions

**Comments to the Author**

1. If the authors have adequately addressed your comments raised in a previous round of review and you feel that this manuscript is now acceptable for publication, you may indicate that here to bypass the “Comments to the Author” section, enter your conflict of interest statement in the “Confidential to Editor” section, and submit your "Accept" recommendation.

Reviewer #1: All comments have been addressed

Reviewer #2: All comments have been addressed

2. Is the manuscript technically sound, and do the data support the conclusions?

Reviewer #1: Yes

Reviewer #2: Yes

3. Has the statistical analysis been performed appropriately and rigorously? 

Reviewer #1: Yes

Reviewer #2: Yes

4. Have the authors made all data underlying the findings in their manuscript fully available?

Reviewer #1: Yes

Reviewer #2: Yes

5. Is the manuscript presented in an intelligible fashion and written in standard English?

Reviewer #1: Yes

Reviewer #2: Yes

6. Review Comments to the Author

Reviewer #1: The quality of this paper has increased significantly after addressing the reviewers' comments. Thus, it is ready for publication.

Reviewer #2: All comments have been addressed satisfactory. It would be good to:

1) include the relevant theories that bind the variables together.

2) include the labelling of hypotheses in Figure 1.

3) Proofread the manuscript to rectify the error such as "(TAM developed by Davis..." at Line 146.

7. PLOS authors have the option to publish the peer review history of their article (what does this mean?). If published, this will include your full peer review and any attached files.

Reviewer #1: No

Reviewer #2: No

---

## [Editor Report · Acceptance letter]

11 Jan 2025

PONE-D-24-32042R1 

PLOS ONE

Dear Dr. Valencia-Arias, 

I'm pleased to inform you that your manuscript has been deemed suitable for publication in PLOS ONE. Congratulations! Your manuscript is now being handed over to our production team.

Kind regards, 

on behalf of

Dr. Mingming Li 

Academic Editor

PLOS ONE